# The role of hypothyroidism in cirrhosis pathogenesis: A retrospective cohort study and multi-omics integration analysis

Ziyang Yang[1,2☯], Weixuan Liang[1,2☯], Qi Zhang[1,2☯], Can Weng[2], Hao Deng[2], Zhuofeng Wen[2], Jingyi Wu[3], Jingwen Deng[2], Zhixin Xie[2], Yiwei Lin[4], Xiuling Fu[2], Chengxin Gu[1], Tao Yang[1], Hui Yang[1]*, Jiyuan Zhou[1]*

1 Department of Gastroenterology, the Second Affiliated Hospital of Guangzhou Medical University, Guangzhou, China, 2 Guangzhou Medical University, Guangzhou, China, 3 Shantou University Medical College, Shantou, China, 4 The First School of Clinical Medicine, Southern Medical University, Guangzhou, China

☯ These three authors contributed equally to this work.
* yanghui@gzhmu.edu.cn (HY); Corjyzhou03@bjmu.edu.cn (JZ)

## Abstract

### Background

Liver cirrhosis is a progressive chronic disease with high morbidity and mortality, thereby posing a major challenge to global health. Evidence suggests that thyroid dysfunction, particularly hypothyroidism, is linked to liver diseases. Hypothyroidism disrupts metabolism, immune homeostasis, and inflammatory pathways, processes central to cirrhosis pathophysiology. However, its causal role and molecular mechanisms remain unclear.

### Methods

The study initiated by analyzing the association between thyroid dysfunction and cirrhosis through retrospective analysis of longitudinal data obtained from the Medical Information Mart for Intensive Care clinical database. To assess genetic correlation, we applied linkage disequilibrium score regression, followed by bidirectional Mendelian randomization to explore potential causal relationships. Through transcriptome-wide association studies, we identified candidate genes, which were then prioritized using a combination of weighted gene co-expression network analysis and differential gene expression data integration. To interpret the biological relevance of these genes, we conducted functional enrichment analyses. We further explored gene function at the cellular level by leveraging single-cell RNA sequencing (scRNA) to map cell-specific expression patterns, analyze intercellular communication, and simulate gene knockouts. Finally, we performed molecular docking and phenome-wide Mendelian randomization to identify potential therapeutic compounds targeting the prioritized genes.

**Data availability statement:** Publicly available datasets were analyzed in this study, as detailed in S1 and S13 Table.

**Funding:** This work was supported by the Guangdong Basic and Applied Basic Research Foundation (Grant Number 2019A1515110060 to Jiyuan Zhou), the Guangzhou Science and Technology Department-School Joint Project (grant number 2023A03J0417 to Jiyuan Zhou), and the Plan on Enhancing Scientific Research in GMU (Grant Number 02-410-2302035XM to Jiyuan Zhou). The funders had no role in study design, data collection and analysis, decision to publish, or preparation of the manuscript.

**Competing interests:** The authors have declared that no competing interests exist.

## Results

Through a combination of observational and genetic insights, we established a causal relationship between hypothyroidism and cirrhosis, identifying hypothyroidism as a risk factor for cirrhosis. Subsequent multi-omics analyses highlighted HLA-DQA1 and CD27 as potential therapeutic targets. ScRNA revealed key roles of these molecules in macrophages and CD8$^+$T cells, and simulated knockouts confirmed their importance in T cell activation and lymphocyte proliferation. Finally, molecular docking analysis identified glycyrrhizic acid and levothyroxine sodium as candidate drugs targeting HLA-DQA1 and CD27, while phenome-wide Mendelian randomization analysis revealed potential adverse effects associated with these targets.

## Conclusions

This study is the first to reveal a causal relationship between hypothyroidism and cirrhosis, potentially driven by immune dysregulation mediated by HLA-DQA1 and CD27. These findings offer novel insights into disease progression and identify HLA-DQA1 and CD27 as potential therapeutic targets, with glycyrrhizic acid and levothyroxine sodium as promising candidate drugs.

### Author summary

Liver cirrhosis is a life-threatening condition with limited treatment options, and understanding its risk factors is essential for early prevention. In this study, we investigated the causal relationship between hypothyroidism—a disorder in which the thyroid gland fails to produce sufficient hormones—and cirrhosis. Using a combination of retrospective and multi-omics studies, we found that hypothyroidism may increase the risk of developing cirrhosis. We also identified two immune-related molecules, HLA-DQA1 and CD27, that may mediate this effect and serve as potential targets for future therapies. Furthermore, our analysis revealed a set of candidate drugs that could help treat patients with both conditions. These findings highlight an overlooked link between thyroid dysfunction and liver disease and offer new directions for clinical intervention and drug development.

## Introduction

Liver cirrhosis, the 11th leading global cause of mortality [1], represents the terminal stage of chronic liver injury marked by hepatic encephalopathy and portal hypertension [2,3]. While viral hepatitis, alcohol-related liver disease, and non-alcoholic fatty liver disease are primary etiologies [4,5], emerging evidence implicates hypothyroidism as a key modifier of cirrhosis progression. Hypothyroidism, affecting 1–7% of adults depending on age and iodine status [6], is predominantly caused by iodine

deficiency and autoimmune disorders, particularly Hashimoto's thyroiditis (HT) [7]. Critically, hypothyroidism exacerbates hepatic fibrosis [8,9] and metabolic dysfunction [10], establishing a bidirectional thyroid-hepatic axis: thyroid hormones regulate hepatic lipid metabolism by promoting mitochondrial β-oxidation and cholesterol efflux [11,12], while impaired hepatic function reduces peripheral thyroxine (T4) to triiodothyronine (T3) conversion [13], creating a self-reinforcing pathogenic cycle. This pathological synergy is evidenced by the correlation between low serum T3 levels and cirrhosis severity [14], as well as preclinical models demonstrating hypothyroidism-induced hepatic steatosis and fibrosis via disrupted autophagy and lipid turnover [15]. These findings suggested that hypothyroidism may increase the risk of cirrhosis through multiple mechanisms.

Large-scale clinical cohorts provide direct evidence for associations between hypothyroidism and cirrhosis, yet such observational studies cannot elucidate the biological mechanisms driving this association [16,17]. Therefore, to reveal their deeper and more complex interaction mechanisms, integrating multidimensional functional evidence is essential. Fortunately, the emergence of a multi-omics framework provides the robust tools to dissect these complex interactions [18,19]. For instance, genome-wide association studies (GWAS) enable the identification of genetic variants, while leveraging their random allocation effectively mitigates confounding in causal inference [20,21]. Further integrating transcriptome-wide association studies (TWAS), genetic regulation of gene expression is mapped to disease mechanisms [22–24], pinpointing genes linking hypothyroidism to cirrhosis. Moreover, single-cell RNA sequencing (scRNA) analysis reveals their cell-type-specific dynamic expression, interactions and regulatory mechanisms [25,26]. Finally, druggability analysis prioritizes therapeutic targets with clinical translational potential [27].

Therefore, this study aimed to systematically evaluate the causal relationship between hypothyroidism and cirrhosis, identify key genes and pathways mediating this association, elucidate the underlying cell-type-specific molecular mechanisms, and ultimately, to screen for and validate potential therapeutic targets. We initially analyzed clinical cohort data and further evaluated the causal relationship between hypothyroidism and cirrhosis by incorporating GWAS summary statistics from the UK Biobank and the FinnGen Consortium. Subsequently, by integrating plasma expression quantitative trait loci (eQTL) data from the eQTLGen Consortium and applying TWAS under the Omnibus Transcriptome Test using the Expression Reference Summary data (OTTERS) framework, we identified candidate genes potentially mediating the complex molecular interplay between hypothyroidism and cirrhosis. To further screen for potential therapeutic targets and dysregulated pathways, we used bulk RNA sequencing. Building on these genetic and transcriptomic findings, we then delineated disease-specific gene expression patterns using scRNA, reconstructed intercellular communication networks, and, through integration with gene regulatory network (GRN) inference, uncovered disruptions in immune interactions driven by key gene perturbations during the progression of cirrhosis. Finally, molecular docking (MD) and phenome-wide Mendelian randomization (PheW-MR) were conducted to evaluate the druggability and potential side effects associated with the identified therapeutic targets (The detailed study design is illustrated in Fig 1).

## Results

### Hypothyroidism Identified as a risk factor for cirrhosis

To investigate the clinical association between hypothyroidism and cirrhosis, we performed a retrospective analysis. The initial study cohort consisted of 39,126 individuals with available thyroid function data, including 1,360 cirrhotic patients and 37,766 healthy controls. Participants were sequentially excluded due to missing data for liver function tests (LFTs), serum electrolytes, or demographic variables. After applying these exclusion criteria, the final analytical cohort comprised 529 cirrhotic patients and 7,664 healthy controls (Fig 2). The proportions of patients with overt hypothyroidism and subclinical hypothyroidism were significantly greater than those of healthy controls. The median age of the patients was 61.5 years. Baseline characteristics revealed that, compared with healthy individuals, cirrhotic patients had higher rates of hypothyroidism and elevated levels of LFTs. Regarding weight status, the patient

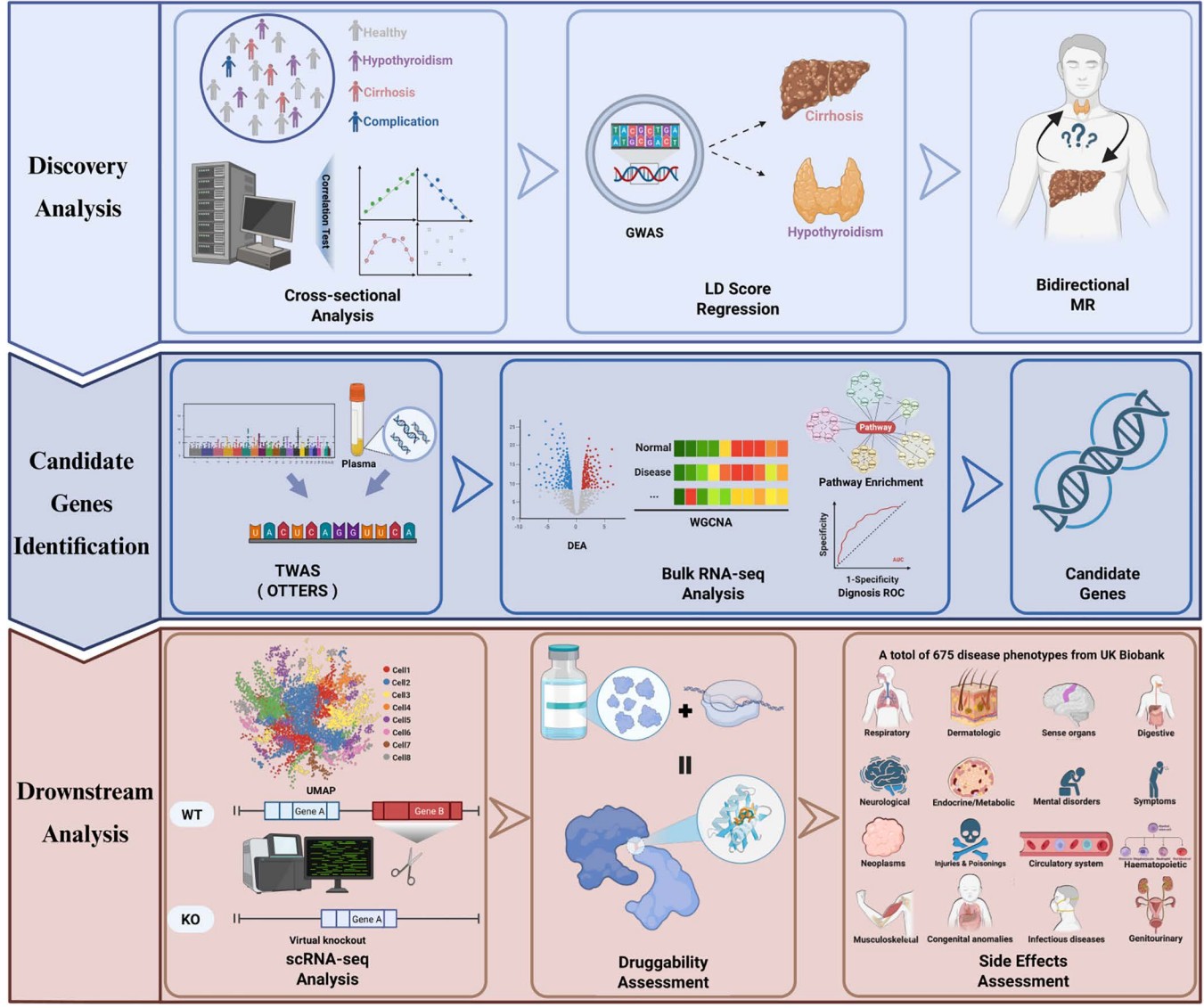

**Fig 1. Schematic representation of the study workflow (Created in BioRender. L, E. (2025)** https://BioRender.com/i3m9apt**).**

exhibited a higher prevalence of overweight and obesity compared with the control group. Univariate logistic regression revealed that overt hypothyroidism ($P = 0.031$, OR [95% CI] = 1.37 [1.02-1.80]) and subclinical hypothyroidism ($P = 0.005$, OR [95% CI] = 1.52 [1.13-2.01]) were significant predictors. After adjusting for demographic data, serum electrolytes, and LFTs in the multivariate logistic regression, this trend remained significant, with overt hypothyroidism ($P = 0.043$, OR [95% CI] = 1.36 [1.01-1.83]) and subclinical hypothyroidism ($P = 0.019$, OR [95% CI] = 1.46 [1.05-1.97]) continuing to be independent risk factors (Table 1).

In this study, the univariable OR provides a crude estimate of the association, while the multivariable OR adjusts for potential confounders including demographic data, serum electrolytes, and LFTs, offering a more accurate effect estimate. An OR > 1 suggests a risk factor, while OR < 1 indicates a possible protective factor. OR = 1 implies no association. A $P < 0.05$ indicates statistical significance.

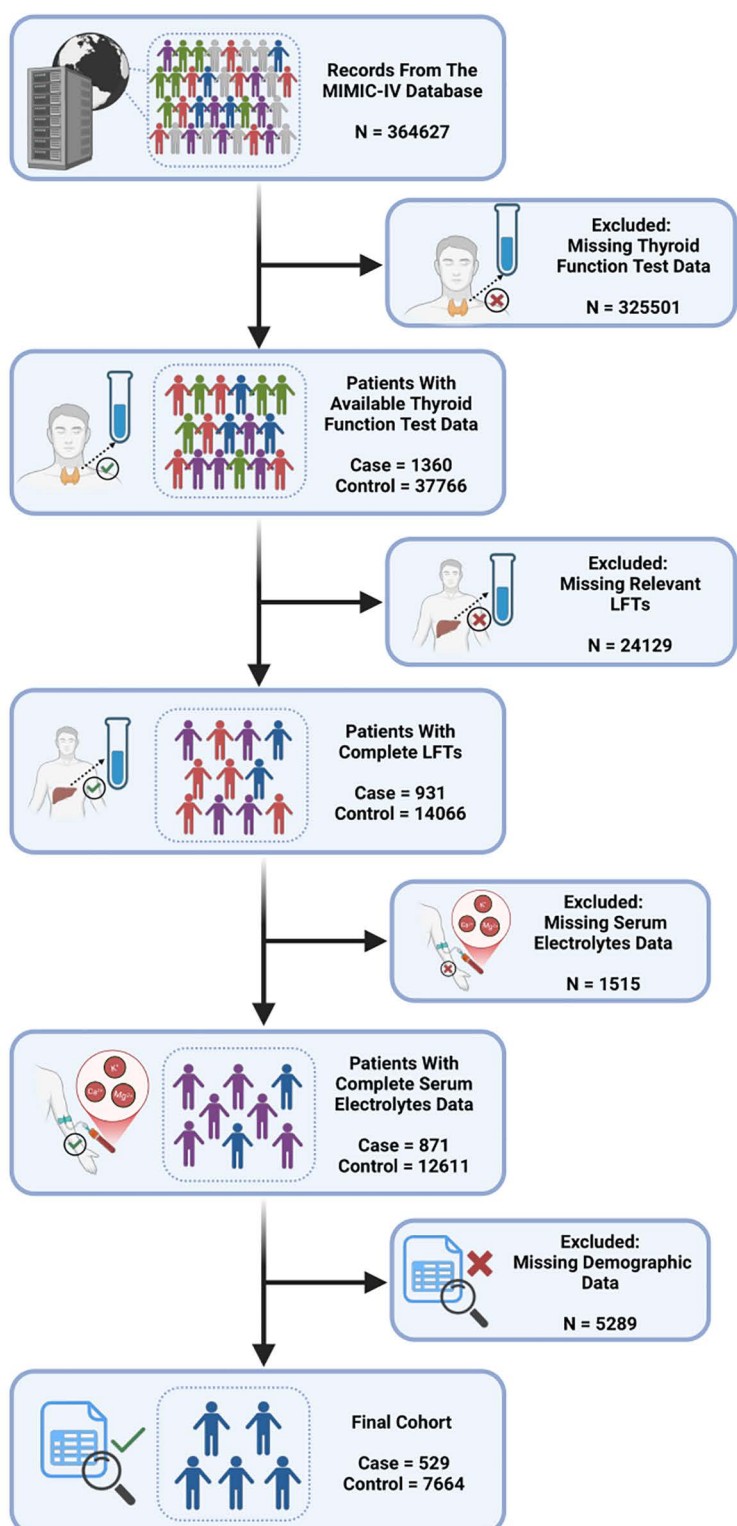

**Fig 2. Flow of included patients through the trial (Created in BioRender. L, E. (2025) https://BioRender.com/minwugm).**

**Table 1. Association Between Thyroid Dysfunction and Cirrhosis Through Univariable and Multivariable Logistic Regression Using MIMIC-IV Data.**

| Characteristic | Description | Control (N = 7, 664) | Case (N = 529) | OR (Univariable) | | OR (Multivariable) | |
|---|---|---|---|---|---|---|---|
| … | … | … | … | OR (95%CI) | P | OR (95%CI) | P |
| Thyroid status | Normal Thyroid Function | 5, 990 (78%) | 391 (74%) | … | … | … | … |
| … | Overt Hypothyroidism | 673 (8.8%) | 60 (11%) | 1.37 (1.02, 1.80) | 0.031 | 1.36 (1.01, 1.83) | 0.043 |
| … | Subclinical Hypothyroidism | 586 (7.6%) | 58 (11%) | 1.52 (1.13, 2.01) | 0.005 | 1.46 (1.05, 1.97) | 0.019 |
| … | Subclinical Hyperthyroidism | 410 (5.3%) | 20 (3.8%) | 0.75 (0.46, 1.15) | 0.215 | 0.74 (0.42, 1.20) | 0.247 |
| … | Overt Hyperthyroidism | 5 (<0.1%) | 0 (0%) | 0.00 | 0.964 | 0.00 | 0.963 |
| Age | Mean ± SD | 66.4 ± 16.3 | 61.5 ± 12.9 | … | … | 0.99 (0.98, 0.99) | <0.001 |
| Gender | Female | 3, 770 (49%) | 244 (46%) | … | … | … | … |
| … | Male | 3, 894 (51%) | 285 (54%) | … | … | 0.95 (0.79, 1.16) | 0.625 |
| Weight status | Normal | 2, 290 (30%) | 117 (22%) | … | … | … | … |
| … | Obesity | 2, 653 (35%) | 212 (40%) | … | … | 1.53 (1.18, 1.98) | 0.001 |
| … | Overweight | 2, 481 (32%) | 191 (37%) | … | … | 1.39 (1.07, 1.82) | 0.016 |
| … | Underweight | 240 (3.1%) | 9 (1.7%) | … | … | 0.47 (0.14, 1.15) | 0.146 |
| Calcium | Mean ± SD | 8.8 ± 0.8 | 8.6 ± 0.8 | … | … | 0.76 (0.67, 0.86) | <0.001 |
| Magnesium | Mean ± SD | 1.6 ± 0.3 | 1.6 ± 0.3 | … | … | 0.91 (0.64, 1.30) | 0.619 |
| Potassium | Mean ± SD | 4.1 ± 0.6 | 4.1 ± 0.6 | … | … | 1.18 (1.00, 1.39) | 0.056 |
| ALT | Mean ± SD | 64.3 ± 241.3 | 67.5 ± 238.2 | … | … | 1.00 (1.00, 1.00) | 0.067 |
| AST | Mean ± SD | 82.5 ± 349.6 | 95.1 ± 260.0 | … | … | 1.00 (1.00, 1.00) | 0.719 |
| TBIL | Mean ± SD | 1.1 ± 2.7 | 3.9 ± 5.9 | … | … | 1.11 (1.09, 1.13) | <0.001 |
| ALP | Mean ± SD | 118.9 ± 142.4 | 162.2 ± 136.7 | … | … | 1.00 (1.00, 1.00) | 0.047 |
| INR | Mean ± SD | 1.4 ± 0.7 | 1.7 ± 0.6 | … | … | 1.40 (1.07, 1.91) | 0.022 |
| PT | Mean ± SD | 15.9 ± 8.2 | 18.5 ± 7.6 | … | … | 0.99 (0.96, 1.01) | 0.382 |
| PTT | Mean ± SD | 33.8 ± 11.1 | 37.6 ± 10.8 | … | … | 1.01 (1.00, 1.02) | 0.001 |

Abbreviations: MIMIC: Medical Information Mart for Intensive Care; OR: Odds Ratio; ALT: Alanine Aminotransferase; AST: Aspartate Aminotransferase; TBIL: Total Bilirubin; ALP: Alkaline Phosphatase; INR: International Normalized Ratio; PT: Prothrombin Time; PTT: Partial Thromboplastin Time; CI: Confidence Interval.

## Genetic correlation and shared genetic architecture between hypothyroidism and cirrhosis

To investigate whether the observed clinical association was underpinned by a shared genetic architecture, we first used Linkage Disequilibrium Score Regression (LDSC) to estimate the genetic correlation between the two diseases. LDSC analysis revealed a significant positive genetic correlation ($P=4 \times 10^{-4}$, rg = 0.2135) between hypothyroidism and cirrhosis, indicating shared genetic risk factors. Furthermore, SNP-based heritability, transformed to the liability scale, was found to be substantial for both diseases. The heritability was estimated at 17.64% (SE = 1.48%) for hypothyroidism and 12.93% (SE = 2.10%) for cirrhosis, with $P$ less than $1 \times 10^{-4}$ for both. These findings suggest that genetic factors play a substantial role in the etiology of both conditions (S2 Table).

## Bidirectional MR revealed a causal effect of hypothyroidism on cirrhosis

Given the observed genetic correlation, we proceeded to investigate the causal relationship between hypothyroidism and cirrhosis using a comprehensive bidirectional MR framework with discovery and replication cohorts. The primary forward MR analysis, using the discovery cohorts for both the exposure and outcome, demonstrated a significant causal association between hypothyroidism and an increased risk of cirrhosis in fixed model ($P=9.7 \times 10^{-4}$, OR [95% CI] = 1.06 [1.02–1.09]). This analysis showed no evidence of both directional horizontal pleiotropy ($P=0.81$) and heterogeneity ($P=0.35$). To rigorously assess the robustness of this finding, we conducted three additional analyses using different combinations of discovery and replication datasets. These analyses largely supported our primary conclusion, showing a consistent direction of effect. Specifically, two of them also yielded statistically significant associations ($P=3.6 \times 10^{-3}$, OR [95% CI] = 1.06 [1.02–1.09]; $P=8.4 \times 10^{-3}$, OR [95% CI] = 1.08 [1.02–1.15]). Although one of the finding, utilizing the datasets both from Finngen consortium, did not reach statistical significance ($P=8.4 \times 10^{-2}$, OR [95% CI] = 1.05 [0.99–1.11]), the effect was directionally consistent with our primary result. Crucially, to synthesize the evidence across all four findings, a meta-analysis was conducted. This provided a robust summary estimate, yielding strong and consistent evidence for a causal link between hypothyroidism and an increased risk of cirrhosis ($P<1 \times 10^{-4}$, OR [95% CI] = 1.06 [1.04–1.08]).

In contrast, the reverse MR analysis revealed no evidence of a causal effect of cirrhosis on hypothyroidism. The results were consistently non-significant across all four analytical combinations (S5 Table, Fig 3A).

## TWAS identified shared risk genes between hypothyroidism and cirrhosis

To pinpoint genes underlying the shared genetic susceptibility between hypothyroidism and cirrhosis, we employed a two-step TWAS strategy. In the first step, TWAS was conducted using the hypothyroidism dataset, resulting in the identification of 292 genes significantly associated with the condition ($P_{FDR}<0.05$). These genes were subsequently evaluated in a second TWAS, this time using the cirrhosis dataset from European ancestry cohorts, to detect overlapping genetic signals. This analysis yielded 46 genes that remained significant ($P_{FDR}<0.05$), suggesting a potential role in the common genetic architecture of both diseases (S6 and S7 Table). A subset of the most strongly associated genes is illustrated in Fig 3B.

## Key immune pathways and hub genes identified by bulk RNA-seq in hypothyroidism and cirrhosis

Building on the genetic findings, we further explored the molecular pathways and functional gene modules that potentially connect hypothyroidism and cirrhosis. Differential expression analysis revealed 516 genes associated with hypothyroidism (DEG-HT) and 1,313 genes related to cirrhosis (DEG-LC), as shown in the volcano plot (Fig 4A). Using Weighted Gene Co-expression Network Analysis (WGCNA), we identified five distinct gene modules in the hypothyroidism dataset (Fig 4B), with the blue module (WGCNA-HT, containing 4,393 genes) exhibiting the strongest positive correlation with the disease phenotype ($P=2 \times 10^{-2}$, r = 0.59) (Fig 4C). Similarly, results of the cirrhosis dataset revealed six gene modules (Fig 4D), among which the turquoise module (WGCNA-LC, 4,629 genes) showed the most robust positive correlation with the phenotype ($P=2 \times 10^{-35}$, r = 0.94) (Fig 4E). The intersection of the DEG discovery set and the WGCNA validation set

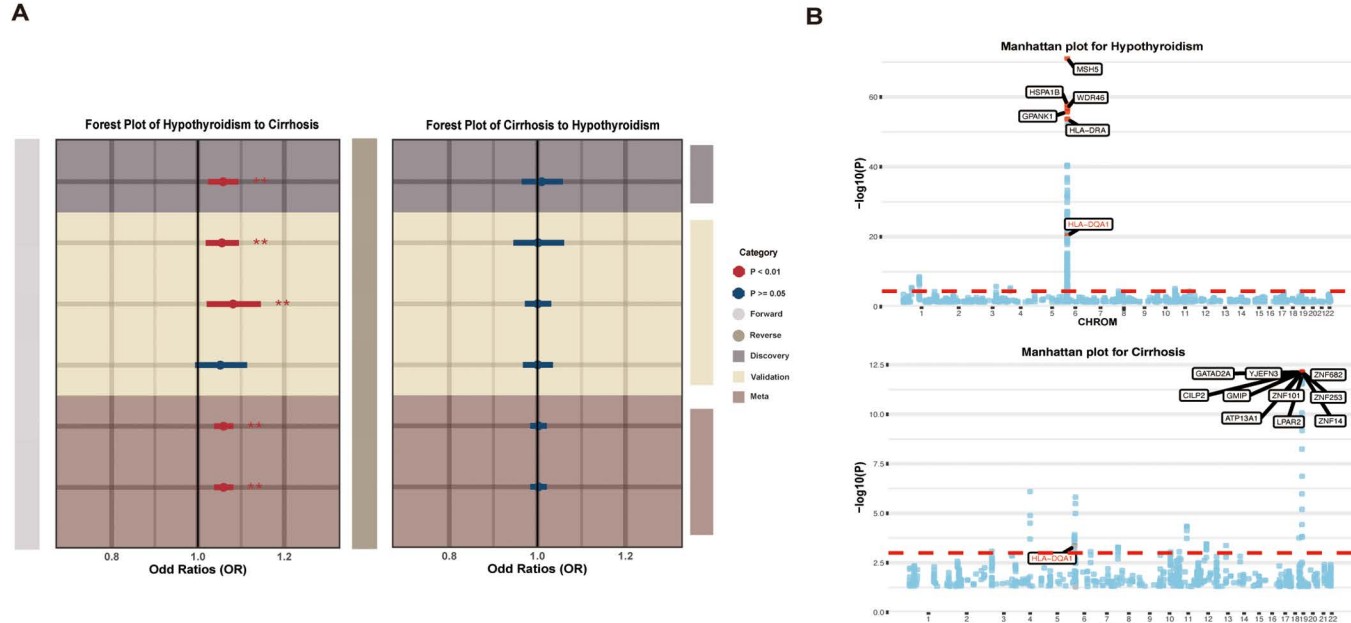

**Fig 3. Mendelian Randomization and Transcriptome-Wide Association Study (TWAS) Analyses.** (A) Forest plot of the bidirectional MR analysis assessing the causal relationship between hypothyroidism and cirrhosis, showing odds ratios (ORs) with 95% confidence intervals from discovery, validation, and meta analyses. (B) Manhattan plots showing TWAS results for significant gene associations. The dotted red line representing $P_{FDR}=0.05$.

yielded key biological pathways, including biological process (BP) terms such as positive regulation of T-cell activation and positive regulation of lymphocyte activation; cellular component (CC) terms including MHC protein, MHC class II protein complex; and molecular function (MF) terms like MHC class II receptor activity and peptide antigen binding. The enriched Kyoto Encyclopedia of Genes and Genomes (KEGG) pathways for these genes included inflammatory bowel disease and primary immunodeficiency (Fig 4F).

By intersecting the TWAS candidate genes, DEGs, and WGCNA module genes, we identified HLA-DQA1 as a core gene (Fig 5A, S8 Table). Further correlation analysis between HLA-DQA1 and the gene sets linked to the top 10 key biological pathways revealed that CD27 stood out as the only gene showing both a strong correlation with HLA-DQA1 and consistent expression patterns across hypothyroidism and cirrhosis (Fig 5B). Based on these results, HLA-DQA1 and CD27 were designated as hub genes. To assess their diagnostic utility, we performed receiver operating characteristic (ROC) curve analysis. The analysis revealed that both genes demonstrated strong diagnostic potential. Specifically, in cirrhosis, HLA-DQA1 exhibited an outstanding area under the curve (AUC) of 0.973, while CD27 also showed robust performance with an AUC of 0.846. In hypothyroidism, HLA-DQA1 maintained high diagnostic accuracy with an AUC of 0.924, whereas CD27, though more moderate, still achieved an AUC of 0.664. These findings highlight the strong potential of HLA-DQA1 and CD27 as biomarkers for both conditions (Fig 5C).

### Immune landscape and virtual knockout analysis reveal key drivers of cirrhosis progression

To resolve immune mechanisms underlying cirrhosis progression, we analyzed cell-type-specific gene expression dynamics. After quality control, 67,285 cells were partitioned into 21 clusters (Fig 6A). Using canonical markers (Fig 6B), these clusters were annotated into 9 cell types (Fig 6C). Expression analysis revealed significant inter-group differences in hub genes. CD27 expression increased in T, B, and natural killer (NK) immune cells as the disease progressed. Moreover, we

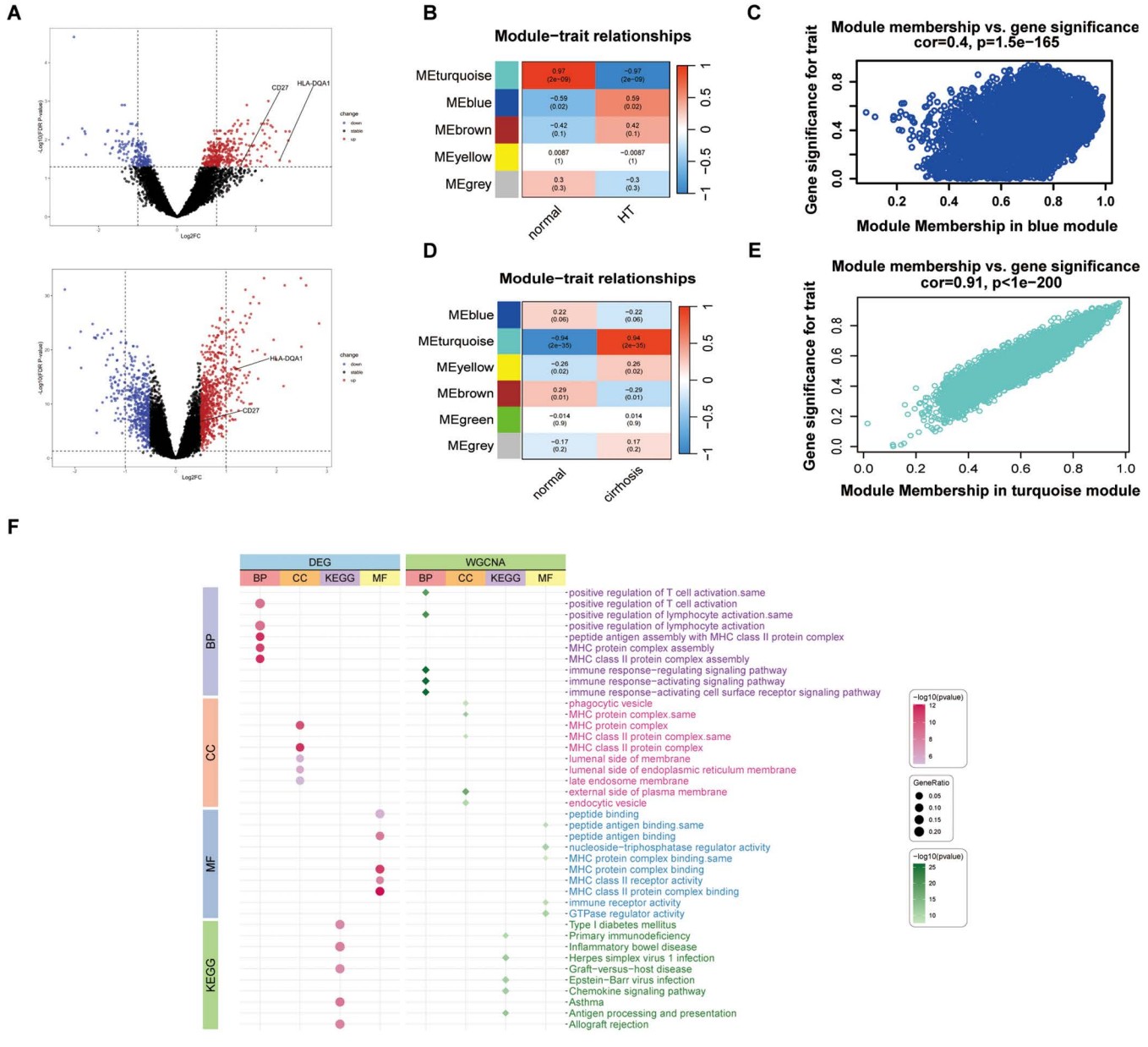

**Fig 4. Differential expression gene (DEG) volcano plots and the results of Weighted Gene Co-expression Network Analysis (WGCNA) and functional enrichment analysis.** (A) Volcano plot for DEGs in hypothyroidism (upper) and liver cirrhosis (lower) indicated that HLA-DQA1 and CD27 were highly expressed in hypothyroidism. (B) Heatmap showing the correlation between module eigengenes and hypothyroidism, with blue indicating negative correlation and red indicating positive correlation. The blue module exhibited the highest positive correlation with hypothyroidism. (C) Module membership and gene significance scatter plot of the blue module for hypothyroidism. (D) Heatmap showing the correlation between module eigengenes and liver cirrhosis, with blue indicating negative correlation and red indicating positive correlation. The turquoise module exhibited the highest positive correlation with liver cirrhosis. (E) Module membership and gene significance scatter plot of the turquoise module for liver cirrhosis. (F) Bubble chart of the functional enrichment analysis for DEGs and module genes, including Biological Process (BP), Cellular Component (CC), Molecular Function (MF), and Kyoto Encyclopedia of Genes and Genomes (KEGG) pathways, was displayed.

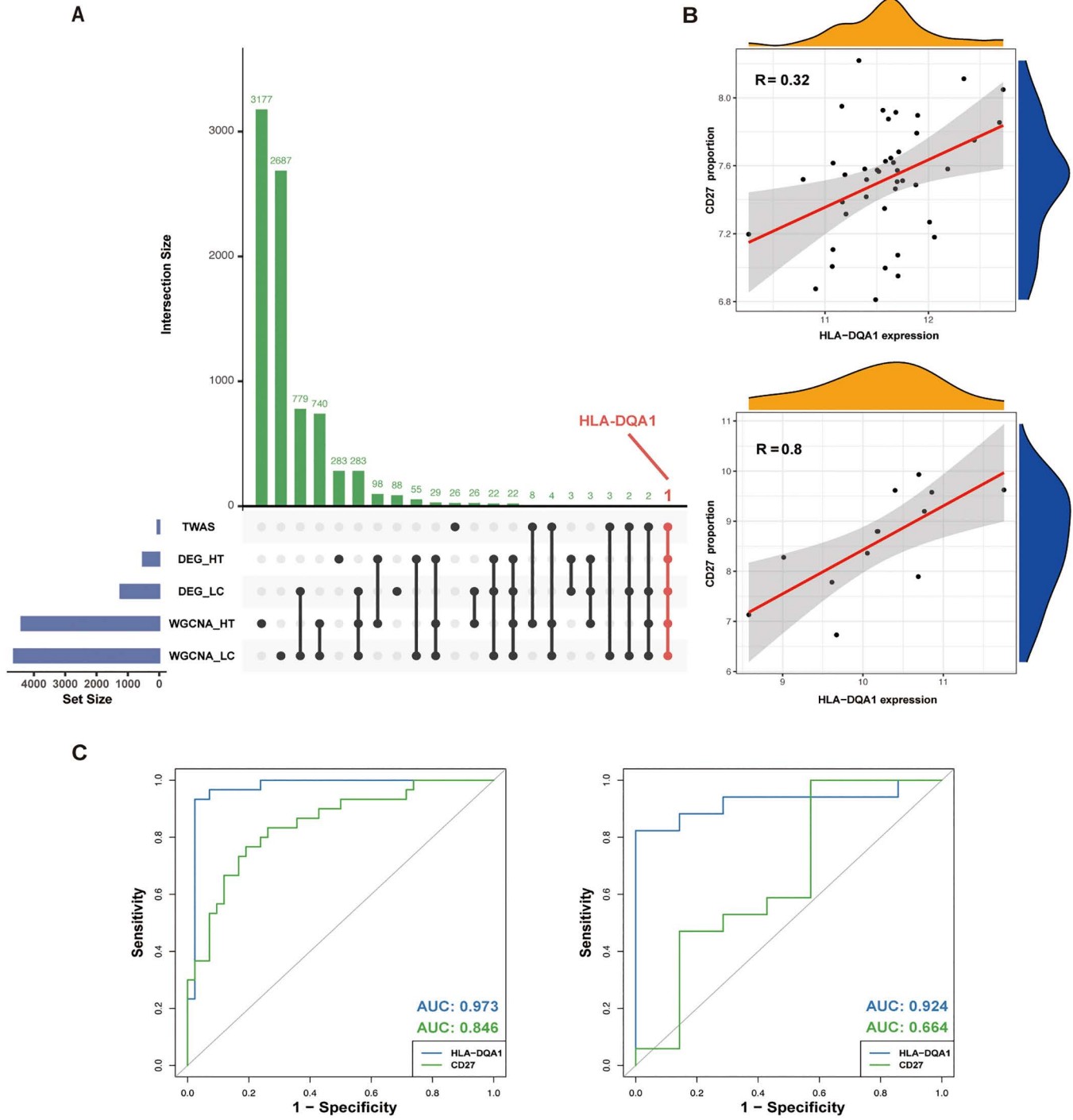

**Fig 5. Identification of Hub Genes, gene correlation and ROC curves for HLA-DQA1 and CD27.** (A) UpSet plot of candidate genes identified by integrating TWAS, DEG-HT, DEG-LC, WGCNA-HT and WGCNA-LC results. The horizontal bar on the left represents several candidate genes obtained from different datasets and methods. Dots and lines represent subsets of genes. Vertical histogram represents number of genes in each subset. Genes identified by integrating TWAS, DEG-HT, DEG-LC, WGCNA-HT and WGCNA-LC results was marked red. (B) Correlation between HLA-DQA1 expression and CD27 proportion in patients with liver cirrhosis (R = 0.32) and hypothyroidism (R = 0.8). (C) The receiver operating characteristic (ROC) of HLA-DQA1 and CD27 in patients with liver cirrhosis (left) and hypothyroidism (right).

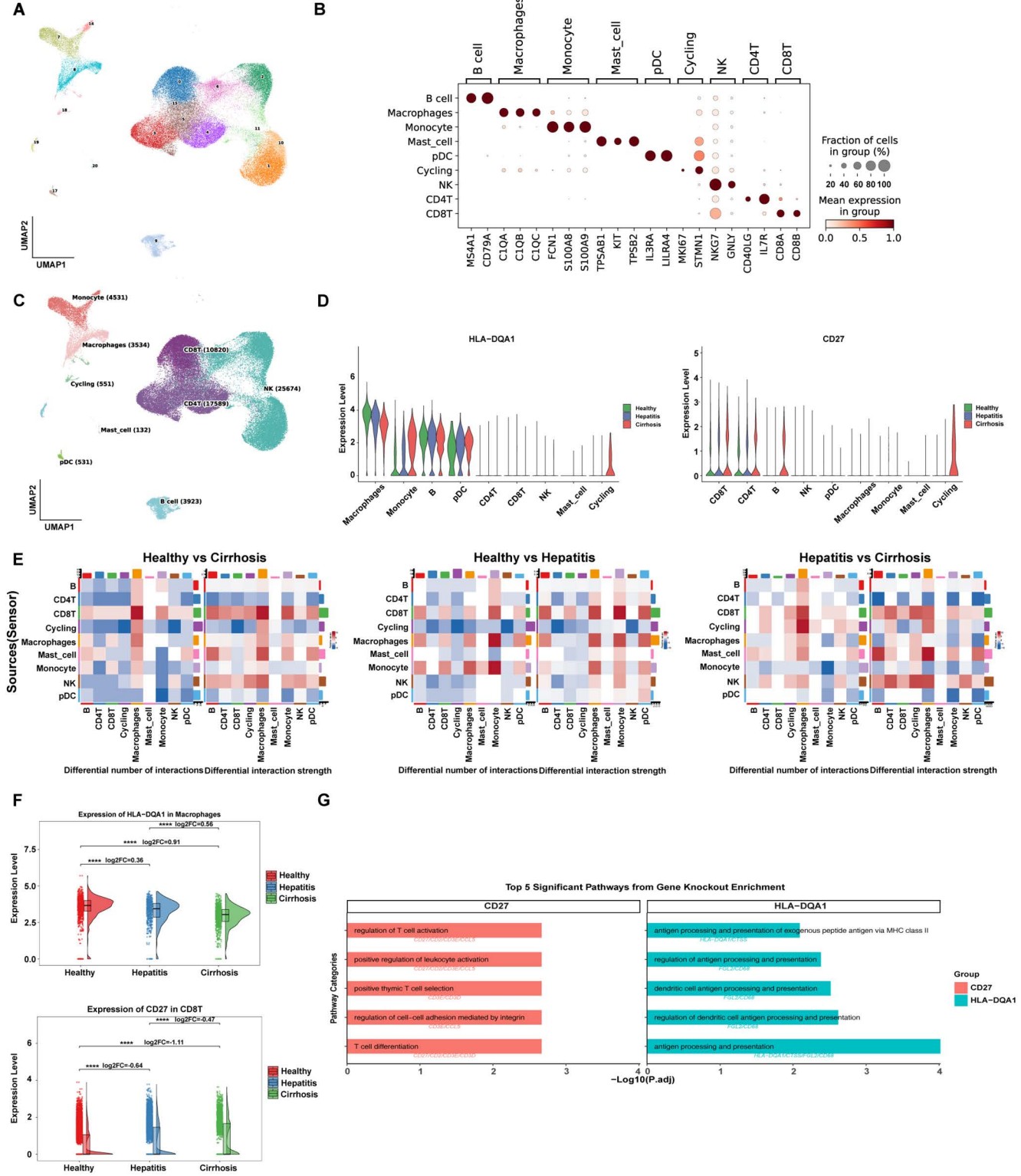

**Fig 6. Results of scRNA-seq analysis.** (A) UMAP plot showing the clustering of cells into 21 distinct clusters based on scRNA. (B) Dot plot displaying significant inter-group differences in hub genes. (C) UMAP plot displaying the distribution of different cell types. (D) Violin plot showing the expression level of HLA-DQA1 and CD27 in different immune cells under different conditions. (E) Communication heatmaps showcasing differences in cell-type

interactions across groups. (F) Violin plots showing the differences in the expression levels of the HLA-DQA1 and CD27 in macrophages among healthy individuals, those with hepatitis, and those with cirrhosis. (G) The bar graph showing the signaling pathways modulated by virtual knockout perturbations.

also found that HLA-DQA1 expression was decreased in macrophages and B cells but increased in monocytes and Plasmacytoid dendritic cells (pDC) (Fig 6D).

Subsequently, cell-cell communication analysis was utilized to reveal distinct interaction networks across the healthy, hepatitis, and cirrhosis groups. Pairwise comparisons via communication heatmaps further showed differences in cell-type interactions across groups. Surprisingly, these comparisons revealed a progressive increase in both the quantity and strength of interactions between CD8+ T cells and macrophages as the disease advanced (Fig 6E). Additionally, we observed that in macrophages, HLA-DQA1 expression progressively decreased with disease progression, whereas in CD8+ T cells, CD27 expression gradually increased. This phenomenon suggests a shift in immune cell activation and interaction patterns, which may reflect the dysregulation of immune responses as the disease progresses, potentially contributing to the progression of cirrhosis (Fig 6F).

Finally, virtual knockout analysis identified 167 genes regulated by HLA-DQA1 and 151 genes influenced by CD27, all with FDR < 0.05 (S10 Table). Pathway enrichment demonstrated that HLA-DQA1 knockout disrupted antigen processing and presentation, regulation of dendritic cell antigen processing and presentation, suggesting its central role in maintaining hepatic immune surveillance. In contrast, CD27 knockout significantly impaired T cell differentiation, and regulation of cell-cell adhesion mediated by integrin, revealing that adaptive immune hyperactivation is mechanistically linked to cirrhosis progression through this hub gene (S11 Table, Fig 6G).

## Multi-target engagement of therapeutic candidates

To evaluate the therapeutic potential of the identified hub genes, we systematically profiled compounds targeting HLA-DQA1 and CD27. Thyroid-related agents exhibited distinct binding patterns: HLA-DQA1 showed strong binding (affinity ≤ -7 kcal/mol) with 31 molecules and moderate binding (-7 to -5 kcal/mol) with 1 molecule, while CD27 interacted strongly with 21 compounds and moderately with 11. Among these, the recently FDA-approved thyroid hormone receptor-beta agonist resmetirom emerged as a dual-target candidate, demonstrating strong binding to HLA-DQA1 (-10.86 kcal/mol, Ki = 10.95 nM) and moderate binding to CD27 (-8.39 kcal/mol, Ki = 709.22 nM). Notably, levothyroxine sodium displayed comparable dual-target capacity, with thyroxine-4-hydroxy-3,5-diiodophenyl ether showing the highest affinity for both HLA-DQA1 (-10.27 kcal/mol, Ki = 29.65 nM) and CD27 (-9.24 kcal/mol, Ki = 168.37 nM). Most strikingly, glycyrrhizic acid, a liver/biliary therapeutic agent, displayed exceptional binding to both targets, with an affinity of -24.5 kcal/mol (Ki = 1.1 aM) for HLA-DQA1 and -18.22 kcal/mol (Ki = 44.11 fM) for CD27, exceeding thyroid-targeted compounds by 1–2 orders of magnitude (S12 Table, Fig 7A-C). These findings underscore the feasibility of therapeutic intervention through multi-target modulation of immune and metabolic pathways.

## PheW-MR revealed side effects of hub genes on disease risk

To evaluate the safety and systemic effects of targeting HLA-DQA1 and CD27, we first performed a PheW-MR analysis across 675 disease traits. After FDR correction, increased expression of HLA-DQA1 was significantly associated with fifteen diseases. Ten of which were linked to reduced disease risk, such as: Gastritis and duodenitis ($P = 1.63 \times 10^{-5}$, OR = 0.93) and Iron-deficiency anaemia of unspecified or non-hemolytic origin ($P = 4.77 \times 10^{-4}$, OR = 0.89). Conversely, HLA-DQA1 was associated with increased risk for five diseases, most prominently Iron metabolism disorders ($P = 1.50 \times 10^{-8}$, OR = 1.60) and Disorders of mineral metabolism ($P = 2.76 \times 10^{-3}$, OR = 1.21) (S14 Table, Fig 7D). Notably, no such associations were found for CD27 as a drug target after correction.

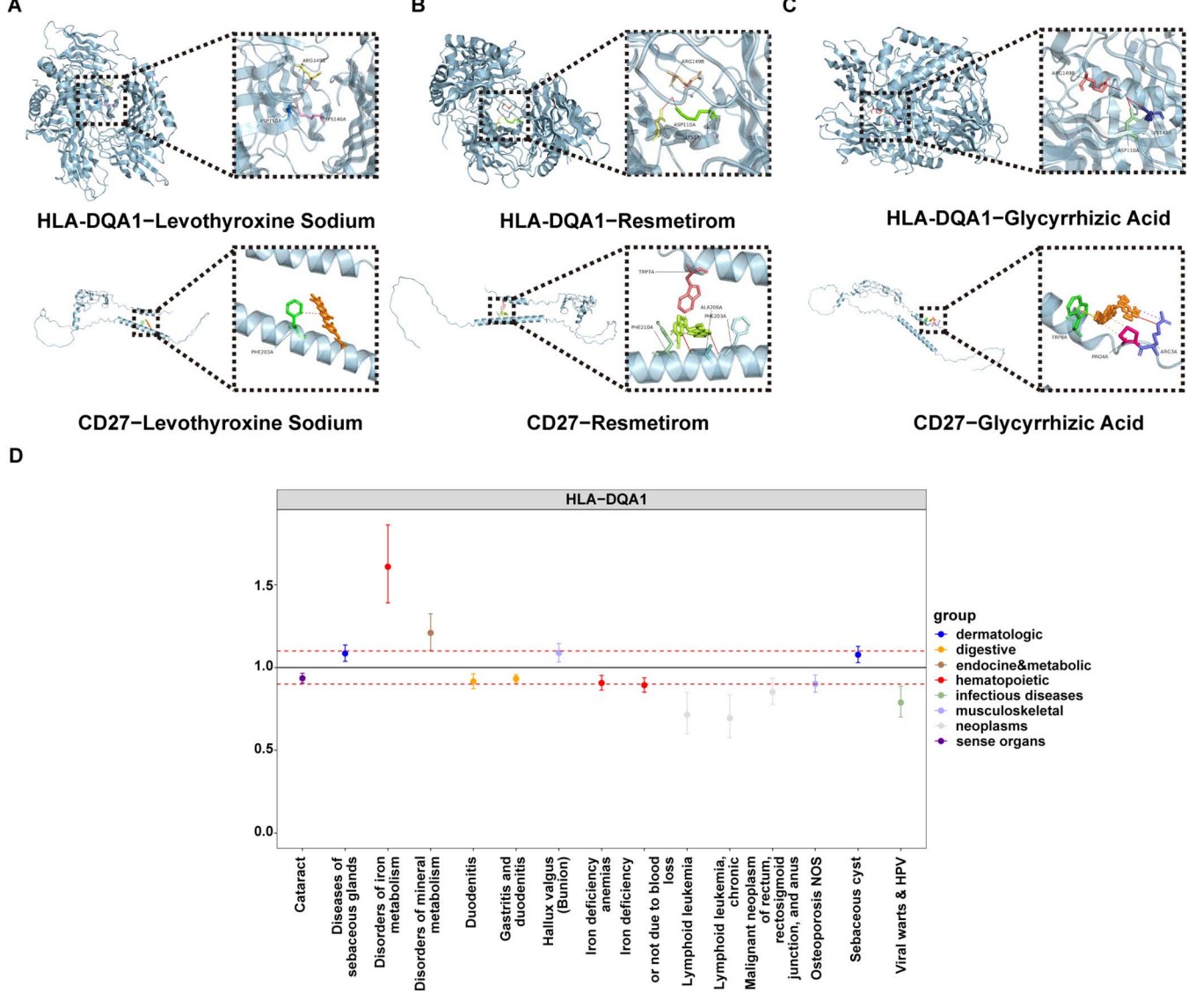

**Fig 7. Results of Molecular Docking analysis and PheW-MR analysis.** (A) to (C) Schematic representation of the docking interactions between HLA-DQA1 and CD27 with glycyrrhizic acid, Resmetirom, and levothyroxine sodium. (D) Forest plot presents the OR (95% CI) for specific disease conditions, with the dotted red line representing the null effect (OR = 1). The color-coded dots correspond to different disease categories, as indicated by the legend on the right. Significant associations are highlighted where the confidence intervals do not cross the null line.

Subsequently, to synthesize the net impact of each gene across all traits, we conducted a random-effects meta-analysis on these results. This revealed that genetically predicted increased expression of HLA-DQA1 was associated with an overall protective effect against the diseases ($P = 1.53 \times 10^{-4}$). In contrast, CD27 showed a non-significant trend towards an overall harmful effect ($P = 0.37$) (S15 Table).

## Discussion

This study provides the first insight into the causal associations, potential biomarkers, and drug targets between hypothyroidism and cirrhosis through clinical cohort and integrated multi-omics approaches. We initially identified a significantly

higher prevalence of both overt and subclinical hypothyroidism among cirrhotic patients compared to healthy controls in a large clinical cohort. This association remained significant even after adjusting for multiple covariates, suggesting that hypothyroidism is an independent risk factor for cirrhosis. Chronic hepatic inflammation and activation of hepatic stellate cells (HSCs) are key events in the pathogenesis of cirrhosis. One of the potential core driver of these processes is the strong association between hypothyroidism and insulin resistance. Thyroid hormone imbalance impairs mitochondrial function and energy metabolism, thereby exacerbating insulin resistance [28,29]. Under insulin-resistant conditions, Kupffer cells release pro-inflammatory cytokines such as Tumor necrosis factor α and Interleukin 6, which subsequently activate HSCs to produce excessive collagen and other extracellular matrix components, accelerating hepatic fibrogenesis and ultimately leading to cirrhosis [30,31]. In addition, hypothyroidism can disrupt the gut microbiota composition and impair intestinal barrier integrity, reduce gastrointestinal motility, and alter luminal potential of hydrogen (pH). These disturbances increase intestinal permeability and facilitate the translocation of bacterial endotoxins into the liver via the portal vein. This process activates Toll-like receptor 4 and upregulates the NF-κB signaling pathway, further promoting the risk of cirrhosis [32–34].

In subsequent genetic analyses, we obtained results consistent with those of observational studies and further established a causal relationship between hypothyroidism and cirrhosis. Genetic correlation analysis revealed a statistically significant yet moderate positive correlation between them, suggesting that genetic factors play a key role in their association. While the risk of cirrhosis is strongly driven by environmental and lifestyle factors, genetic susceptibility is increasingly recognized as a crucial determinant of an individual's risk [35]. Against this backdrop, hypothyroidism has been epidemiologically and clinically associated with various chronic liver diseases that serve as major precursors to cirrhosis. For instance, a notable genetic overlap has been established between autoimmune thyroid disease and primary biliary cholangitis (PBC), a progressive autoimmune liver disease that culminates in cirrhosis [36]. Similarly, recent large-scale analyses have reported a significant positive genetic correlation between hypothyroidism and metabolic dysfunction-associated steatotic liver disease (MASLD), the most common antecedent of cirrhosis [37]. To our knowledge, our study is the first to formally propose this shared genetic basis at the level of cirrhosis itself. This raises the intriguing possibility that a shared genetic predisposition with hypothyroidism might not only be involved in the initiation of specific liver conditions but could also contribute to the overall risk of developing cirrhosis.

To progress from genetic association to causal inference, MR was applied. More importantly, it revealed a unidirectional causal effect of hypothyroidism on the risk of developing cirrhosis. This causal inference implies that hypothyroidism may actively contribute to the of cirrhosis, rather than merely representing a secondary manifestation of impaired liver function. Supporting this, previous studies have demonstrated that functional genes mediating the effects of thyroid hormones, which are essential for regulating fatty acid β-oxidation, also participate in hepatic lipid metabolism [38,39]. Furthermore, in hypothyroid states, reduced T3 levels suppress fatty acid oxidation and cholesterol clearance pathways in hepatocytes, leading to lipid accumulation. Such metabolic dysregulation contributes to the onset of hepatic steatosis and its progression toward fibrosis [37,40]. These findings not only reinforce our research's results but also highlight the potential for clinical translation. By identifying hypothyroidism as an upstream risk factor in the pathogenesis of cirrhosis, our results suggest that early control of thyroid dysfunction may provide a novel avenue for the prevention and treatment of liver cirrhosis.

Notably, our study revealed that HLA-DQA1 may play a significant role in the process by which hypothyroidism promotes the risk of cirrhosis. In hypothyroidism, recent studies have revealed that immune-related genetic variants, such as HLA-DQA1_rs17426593, are closely associated with disease occurrence, which is consistent with the primary cause of hypothyroidism- autoimmune disease (such as Hashimoto's thyroiditis) [41]. The persistent autoimmune destruction of thyroid tissue not only affects the production of thyroid hormones (T3 and T4) but also may influence the liver through HLA-DQA1-mediated immune responses. In cirrhosis research, MHC class II alleles are associated with various forms of cirrhosis, particularly viral hepatitis and primary biliary cholangitis [42,43]. Notably, our scRNA results revealed a novel

finding: as cirrhosis progresses, HLA-DQA1 expression in macrophages tends to decrease. This change in expression pattern suggests that hypothyroidism may participate in the development of cirrhosis by affecting HLA-DQA1 expression and altering the antigen presentation function of macrophages. Additionally, we observed enhanced interactions between CD8+ T cells and macrophages as the disease progressed, which may be a compensatory immune response triggered by HLA-DQA1 downregulation, further promoting liver inflammation and fibrosis.

In addition to HLA-DQA1, our study identified CD27 as another key molecule linking hypothyroidism and cirrhosis. As a member of the tumor necrosis factor receptor superfamily, CD27 plays crucial roles in the pathogenesis of both autoimmune thyroid disease (AITD) and cirrhosis [44]. In AITD patients, CD27 expression is significantly elevated, and the level of its soluble form (sCD27) is positively correlated with anti-thyroid antibody levels. Notably, the increase in CD27+ memory B cells promotes the production of high-affinity autoantibodies, exacerbating thyroid autoimmune damage [44]. During liver disease progression, spatial transcriptomics and multiplex immunofluorescence show the aggregation of CD27+ memory B cells and plasma cells in the portal tract of PBC patients [45]. Our scRNA data further confirmed this finding: as cirrhosis progresses, CD27 expression levels in CD8+ T cells increase, closely correlating with enhanced interactions between CD8+ T cells and macrophages. These findings suggest that hypothyroidism may initially activate autoimmune responses by increasing CD27 expression, and subsequently, this immune imbalance promotes liver inflammation and fibrosis progression through sustained activation of the CD27/CD70 costimulatory pathway.

To explore the potential complex interactions between these therapeutic targets and various drugs, we employed MD techniques and demonstrated significant binding affinities of glycyrrhizic acid, levothyroxine sodium, and Resmetirom to both HLA-DQA1 and CD27. Glycyrrhizic acid, a natural compound extensively employed in hepatic therapeutics, exerts anti-inflammatory and antioxidant effects through modulation of signaling pathways such as Nuclear Factor kappa-light-chain-enhancer of activated B cells [46,47]. Extensive clinical data indicate that excessive thyroid hormone replacement therapy for hypothyroidism may exacerbate metabolic stress-induced liver injury [48–50]. However, it is noteworthy that levothyroxine sodium, the first-line therapy for hypothyroidism, rarely exhibited hepatotoxicity [51]. Our findings propose a novel therapeutic strategy for preventing cirrhosis development in hypothyroidism patients. Future drug development or clinical interventions could combine levothyroxine sodium as a hormonal supplement with the hepatoprotective effects of glycyrrhizic acid, thereby simultaneously addressing thyroid dysfunction and mitigating the risk of hepatic injury and cirrhosis. Moreover, resmetirom, a novel therapeutic agent for Non-Alcoholic Steatohepatitis, exerts its effects by selectively activating THR-β [52]. Following hepatocyte-specific uptake, it mitigates hepatic lipid accumulation, inflammation, and fibrosis through multiple pathways [53]. Notably, its strong binding affinity to hub genes suggests a potential regulatory mechanism independent of THR-β in the treatment of liver fibrosis secondary to thyroid dysfunction. This mechanism may slow the progression of cirrhosis, offering a novel therapeutic approach for hypothyroidism-related liver fibrosis.

In the present study, we evaluated the potential side effects of therapeutic approaches targeting HLA-DQA1 and CD27 through PheW-MR analysis. The results of the drug safety assessment revealed the complexity of HLA-DQA1 as a therapeutic target. Its bidirectional regulatory effects on multiple disease phenotypes reflect its broad functionality in the immune system. As a major histocompatibility complex class II molecule, HLA-DQA1 plays a central role in antigen presentation and immune responses, and this universality means that therapeutic interventions targeting it may have systemic effects [44]. In particular, its significant association with iron metabolism disorders suggests potential interactions between the immune system and iron homeostasis regulation. This finding is consistent with previous reports of iron metabolism abnormalities in autoimmune disease patients [54]. Notably, the protective effects of HLA-DQA1 against gastroduodenitis and iron deficiency anaemia appear to contrast with its risk for iron metabolism disorders, a seemingly paradoxical phenomenon that may reflect compensatory regulatory mechanisms of the immune system in different pathological processes. In contrast, the expression and function of CD27, a costimulatory molecule, are limited to specific immune cell subsets, particularly T cells. This cell-type specificity may explain why CD27-targeted therapeutic strategies

are associated with lower risks of systemic adverse reactions, echoing recent experiences in cancer immunotherapy, where targeting costimulatory/coinhibitory molecules typically results in manageable safety profiles.

Compared to traditional approaches, our study offers several distinct advantages. First, by integrating findings from both observational and genetic studies, we systematically revealed for the first time the complex causal relationship between hypothyroidism and cirrhosis. Second, in contrast to conventional TWAS framework, we employed an innovative TWAS method based on the OTTERS framework to identify key plasma proteins linking hypothyroidism and cirrhosis. A key strength of this approach lies in its integration of multiple statistical models to calculate weights for each eQTL, allowing for the efficient utilization of summary-level data and yielding results with enhanced robustness and reliability [55]. Third, rather than relying on single-omics strategies, we adopted a multi-omics approach that integrated genomic and transcriptomic data for gene identification and validation. This provided a more comprehensive understanding of gene–disease relationships. Fourth, while traditional genetic analyses often neglect the biological mechanisms underlying disease progression, our study combined genetic data with scRNA and enrichment analysis to explore the functional roles of candidate genes in the development of cirrhosis. Finally, building on current clinical treatment practices, we investigated the druggability of potential therapeutic targets and evaluated their possible side effects. This not only expands the translational value of our findings but also lays the groundwork for future therapeutic interventions.

Despite yielding several important findings, our study is not without limitations. First, our research was primarily based on European population data, which may limit the generalizability of the results to other ethnic groups. that hypothyroidism and cirrhosis may have different disease characteristics and risk factors in Asian populations, future validation in diverse populations is needed. Second, because the control group was less likely than the case group to undergo relevant clinical examinations during treatment, our retrospective analysis resulted in substantial missing data. This differential missingness may have, in turn, introduced selection bias; consequently, our observational findings warrant validation in future prospective studies. Third, in the transcriptome data analysis, owing to the lack of bulk RNA-seq data for hypothyroidism, we used HT data as a substitute. Although HT is one of the most common causes of hypothyroidism, its distinct immunological characteristics might lead to an overemphasis on autoimmune-specific pathways, potentially conflating them with the features of hypothyroidism itself. Therefore, future studies on transcriptomic data from patients with well-defined hypothyroidism are necessary to validate our findings. Finally, our study primarily focused on the immunological mechanisms of the hypothyroidism-cirrhosis relationship, while other potential mechanisms, such as metabolic disorders and hormonal influences, require further investigation. Considering the important role of thyroid hormones in liver metabolism, integrating metabolomics data and thyroid hormone signaling pathway analysis may provide additional insights into disease mechanisms.

## Materials and methods

### Data source

**MIMIC-IV cohort.** This retrospective cohort analysis used the MIMIC-IV database [16], which includes data from 364,627 patients at Beth Israel Deaconess Medical Center (2008–2022). The database contains patient demographics, laboratory results, vital signs, and prescriptions. The datasets are publicly available, and the personal privacy information of patients in this database is de-identified.

**GWAS resources.** GWAS summary statistics for hypothyroidism and cirrhosis were obtained from four independent cohorts to support a two-stage discovery and validation design. For the discovery phase, this study utilized data from the UK Biobank for hypothyroidism [56], and a meta-analysis by Ghouse *et al.* for cirrhosis [19]. For subsequent replication, independent GWAS data for both traits were sourced from the FinnGen consortium [57,58] (S1 Table).

**eQTL data.** Cis-eQTL (within ±1 megabase of gene transcription start sites) summary-level data for 16,699 genes were derived from the eQTLGen Consortium, based on a meta-analysis of 31,684 blood-derived samples across 37 cohorts. The dataset involved 25,482 samples derived from whole blood, accounting for 80.4% of the total, and 6,202 samples from peripheral blood mononuclear cells, which represented the remaining 19.6% [59].

**RNA sequencing data.** RNA sequencing datasets were obtained from the Gene Expression Omnibus (GEO) repository. The bulk transcriptomic cohort included 72 liver tissue samples—30 from normal controls and 42 from cirrhotic patients, sourced from GSE89632, GSE164760, and GSE56140. Additionally, 24 thyroid tissue samples were analyzed, consisting of 7 normal controls and 17 HT samples, retrieved from GSE176153 and GSE138198. HT samples served as a phenotypic proxy for hypothyroidism due to the limited availability of bulk RNA-seq datasets explicitly profiling hypothyroidism.

The scRNA data included samples from 4 healthy controls, 3 hepatitis patients and 5 cirrhosis patients from GSE136103 and GSE159977, focusing on immune cell subgroup.

## Thyroid dysfunction assessment and cohort selection in the MIMIC study

This study included all patients with cirrhosis who underwent thyroid function tests, as well as healthy controls. Thyroid dysfunction was determined by measuring Thyroid-Stimulating Hormone (TSH) and total T4 levels [60]. Subclinical hyperthyroidism was defined as a TSH level greater than 0.45 mIU/L, with a normal T4 level or no T4 measurement, as overt hyperthyroidism is rare [60,61]. Overt hypothyroidism was defined as a TSH level ≥ 20 mIU/L, or a TSH level ≥ 4.5 mIU/L with a T4 level below the normal range. Overt hyperthyroidism was defined as a TSH level less than 0.45 mIU/L with a T4 level above the reference range. The reference range for T4 was 4.5 to 13.2 µg/dL. Patients were excluded the following criteria: (1) age < 18 years; (2) liver cancer, pregnant, or breastfeeding; and (3) admitted for less than 24 hours.

Additionally, the analysis included covariates from three main categories: demographic data, namely age, gender, and weight status; serum electrolytes, consisting of calcium, magnesium, and potassium; and LFTs, including alanine aminotransferase (ALT), aspartate aminotransferase (AST), total bilirubin (TBIL), alkaline phosphatase (ALP), international normalized ratio (INR), prothrombin time (PT), and partial thromboplastin time (PTT). Weight status was categorized based on Body Mass Index (BMI) as follows: Underweight (BMI < 18.5 kg/m²), Normal weight (18.5 ≤ BMI < 25.0 kg/m²), Overweight (25.0 ≤ BMI < 30.0 kg/m²), and Obesity (BMI ≥ 30.0 kg/m²) [62]. For data preprocessing, patients with missing data were excluded from the analysis. Data extraction was performed via PostgreSQL and Navicat Premium 17.

The distribution of continuous variables was assessed using the Shapiro-Wilk test, with the results expressed as means ± SD. Group differences were analysed via the t-test or rank-sum test for continuous variables and the chi-square test for categorical variables, with < 0.05 indicating statistical significance. Univariate logistic regression was performed to assess the relationship between thyroid function and cirrhosis, and all other variables were included as covariates in the multivariate logistic regression analysis.

## Genetic correlation analysis

LDSC was used to estimate the SNP-based heritability ($h^2_{SNP}$) of hypothyroidism and cirrhosis, as well as the genetic correlation ($r_g$) between them. This method links GWAS test statistics to LD and estimates genome-wide heritability while addressing confounding factors such as population stratification and cryptic relatedness [63]. Precomputed LD scores from the 1000 Genomes Project were applied [64]. For these case-control traits, SNP-based heritability was converted from the observed scale to the liability scale using population prevalence estimates: the estimate for cirrhosis was derived from the Global Burden of Disease (GBD) study [65], while the estimate for hypothyroidism was reported by Taylor PN *et al.* [66], as the latter was not available in the GBD.

## Bidirectional Mendelian randomization

This study employed a bidirectional MR approach to explore the causal relationship between hypothyroidism and cirrhosis. The primary analyses was conducted using the discovery cohorts and subsequently performed in the various validation cohorts for both the forward (hypothyroidism to cirrhosis) and reverse (cirrhosis to hypothyroidism) directions. The MR analysis relied on three core assumptions: the relevance assumption requiring a strong association between instrumental

variable (IV) and exposure; the independence assumption ensuring IVs are unaffected by confounders; and the exclusion restriction assumption stipulating that IVs influence the outcome solely through the exposure [67].

Genetic instruments for each direction were selected based on genome-wide significance ($P < 5 \times 10^{-8}$) and refined using LD clumping thresholds ($R^2 = 0.001$, clumping distance = 10,000 kb) to ensure independence. Single Nucleotide Polymorphisms (SNPs) with F-statistics < 10, calculated as $F = \frac{R^2 \times (N-1-k)}{(1-R^2) \times k}$ where $R^2$ is the variance explained by IVs, $N$ is the sample size, and $k$ is the number of IVs, were excluded to minimize weak instrument bias [68]. To further ensure the correct causal direction, we applied Steiger filtering, removing any IVs that explained more variance in the outcome than in the exposure [69]. To rigorously uphold the core assumptions of MR, we coned a comprehensive screening for potential horizontal pleiotropy. First, we directly screened all IVs against the outcome and removed any variants associated with it that surpassed a significance threshold of $P < 1 \times 10^{-5}$. Second, each IV was also systematically queried against the GWAS Catalog (https://www.ebi.ac.uk/gwas/) to identify any previously reported associations with the outcome or its major risk factors using the same stringent threshold. Any IV demonstrating a significant association with these traits was presumed to exhibit horizontal pleiotropy and was consequently removed from the final set of IVs used in our analysis (S3 and S4 Table).

Causal effects were estimated for both directions primarily using the inverse variance weighted (IVW) method; a random-effects model was employed in the presence of significant heterogeneity, otherwise, a fixed-effect model was used [70,71]. And this was supplemented by weighted median, MR-Egger, simple mode, and weighted mode approaches for robustness and sensitivity [20]. Heterogeneity among IVs was assessed via Cochran's Q test ($P < 0.05$ indicating significant heterogeneity) and quantified using the I² statistic. Furthermore, the MR-PRESSO method was used to detect and correct for potential horizontal pleiotropic outliers [72]. The leave-one-out sensitivity analysis was performed to evaluate the influence of individual SNPs on the results [20]. Finally, a meta-analysis was performed to synthesize the causal estimates from the discovery and replication cohorts to derive a summary effect [73,74].

**Transcriptome-wide association study**

We employed the OTTERS framework to conduct two TWAS analyses using GWAS summary statistics for hypothyroidism from UK Biobank cohorts and cirrhosis from Jonas Ghouse *et al* [19,56]. eQTL data from the eQTLGen Consortium were integrated with these GWAS datasets to refine genetic expression profiles and estimate cis-eQTL weights [59]. It operates in two stages [55]. In the first stage, cis-eQTL weights were estimated using four polygenic risk score methods (P+T, lassosum, SDPR, PRS-CS) and combined with LD reference panels to predict genetically regulated expression levels (GReX). In the second stage, imputed GReX values were integrated with GWAS summary statistics to perform gene-based association testing, with $P$ aggregated using the ACAT-O method. FDR correction ($P_{FDR} < 0.05$) was applied to account for multiple testing.

In our study TWAS was performed in two-step, firstly, hypothyroidism GWAS summary statistics were used to identify significantly associated genes ($P_{FDR} < 0.05$). Subsequently we reanalyzed these genes in for cirrhosis to identify shared genetic associations, enabling the discovery of co-morbid genes between the two traits.

**Processing and analysing bulk RNA sequencing data**

Transcriptomic datasets for hypothyroidism and cirrhosis patients underwent standardized preprocessing including filtering genes with whose average expression below 1, batch effect correction, and normalization. Differential expression analysis identified condition-specific DEGs through linear modeling with empirical Bayesian adjustment, applying thresholds of absolute log2 fold change greater than 0.5 and adjusted $P$ less than 0.05. Intersecting DEGs between both diseases were subjected to Gene Ontology (GO) and KEGG enrichment analysis ($P_{FDR} < 0.05$) to establish discovery pathways [75,76]. To validate whether these pathways are shared between hypothyroidism and cirrhosis, WGCNA—a method that constructs gene co-expression networks by optimizing soft-thresholding power to detect phenotype-associated gene clusters

reflecting coordinated regulatory mechanisms—was employed [77]. A detailed description of the WGCNA workflow, including network construction and module identification, is provided in the S1 Text. The most phenotype-correlated and statistically significant modules in each disease cohort were selected for GO/KEGG enrichment analysis. Pathways overlapping between the discovery and validation sets were defined as key biological pathways potentially shared by both diseases.

The intersection of TWAS candidates, DEGs, and WGCNA module genes yielded a core gene. The genes associated with the top ten key enriched pathways identified through differential analysis were subjected to Spearman correlation analysis, and the significantly co-expressed genes were further selected as hub genes. Diagnostic performance was quantitatively evaluated through ROC curve analysis, with AUC values calculated to assess its discriminative capacity for hypothyroidism and cirrhosis.

## Processing and analysing single-cell RNA sequencing data

scRNA data were processed using the Seurat V5.0 R package [78]. Transcriptional profiling focused on immune cell subsets with stringent quality control: cells retaining 300–5,000 expressed genes, total RNA counts > 500, mitochondrial gene content < 20%, hemoglobin gene expression < 3%, platelet gene expression < 1% were retained. Doublets cells were removed by estimating a 5% doublet probability, and environmental RNA contamination was eliminated by using the decontX R package [79,80]. Data normalization was performed through scaling, followed by identification of 3,000 highly variable genes. Dimensionality reduction and clustering were conducted using principal component analysis. Cell type annotation integrated automated classification using Human Primary Cell Atlas Data as a reference [81], along with manual verification through canonical markers: CD79A for B cells, IL3RA for Plasmacytoid dendritic cells (pDC), KIT for mast cells, C1QA for macrophages, FCN1 for monocytes, CD40LG for CD4+ T cells, CD8A for CD8+ T cells, STMN1 for cycling cells, and KLRF1 for natural killer (NK) cells.

Cell-cell communication analysis was employed to decipher the intercellular signaling dynamics underlying disease progression. Using CellChat, we systematically compared ligand-receptor interaction networks across healthy, hepatitis, and cirrhotic microenvironments, enabling the identification of pathological communication patterns that may drive immune dysregulation in hepatothyroid axis disorders [82].

To investigate the regulatory hierarchy of hub genes during cirrhosis progression, virtual knockout analysis was performed using scTenifoldKnk [83]. This approach constructed denoised single-cell gene regulatory networks (scGRNs) from scRNA-seq data, enabling systematic interrogation of hub gene function through targeted network perturbations. The knockout simulation was implemented by systematically removing all outgoing regulatory edges of target hub genes within the reconstructed scGRN. Directed connections originating from these hub genes were computationally ablated to mimic loss-of-function states. Network perturbation effects were quantified through manifold learning-based dimensional reduction analysis, where genes showing substantial coordinate displacements between wild-type and perturbed networks were identified as significantly regulated targets. Disturbed genes were ranked according to the fold change in projection distance between the two scGRN states, with statistical significance assessed using $X^2$ tests. Genes exhibiting both significant displacement and direct network connections to the target hubs were prioritized. GO enrichment analysis was subsequently conducted on the top 10 most affected genes to delineate impacted biological pathways.

## Molecular docking

MD was performed to evaluate the binding potential of approved drug compounds for thyroid and hepatobiliary with proteins encoded by hub genes, with a focus on binding-free energy. Protein structures were retrieved from the Protein Data Bank (PDB) using GetPDB, and ligand structures were obtained from the ChEMBL and PubChem database [84]. Drugs were selected based on Anatomical Therapeutic Chemical (ATC) classifications, including treatments for liver and biliary diseases (A05), thyroid hormones (T3 and T4), and thyroid preparations (H03A), resulting in a total of 202 small

molecules for analysis. Docking simulations were conducted using Dockey and QuickVina-W [85,86], with binding affinities categorized as strong (≤ -7 kcal/mol), moderate (-7 to -5 kcal/mol), or weak/negligible (> -5 kcal/mol). The results were visualized and analysed using PyMOL.

**PheW-MR analysis in UK Biobank**

PheW-MR analysis was conducted on these proteins in various disease traits following the evaluation of their druggability, in order to comprehensively characterize potential adverse effects linked to each target. SNP data for these targets were sourced from eQTL studies and aligned with datasets used in TWAS analysis. We integrated GWAS summary statistics from a UK Biobank cohort (N ≤ 408,961) provided by Zhou *et al.* [56]. A total of 675 traits were selected from 1,403 binary phenotypes after filtering for traits with more than 500 cases, excluding cirrhosis and hypothyroidism, to assess genetic associations between drug targets and disease risk. Summary-data-based mendelian randomization analysis was then performed across these traits. Diseases were categorized using PheCodes, a system that converts International Classification of Diseases (ICD) codes into phenotypic outcomes, enabling a comprehensive genetic analysis of various disease traits.

Furthermore, to synthesize the total effects for each gene, we conducted a random-effects meta-analysis [56] on the 675 trait-specific results. This approach generated a overall genetic effect for each gene, quantifying its net impact across the diverse disease traits while robustly accounting for potential effect heterogeneity. The detailed statistical model is described in the Supplementary Methods.

# Supporting information

**S1 Table. Summary of GWAS, eQTL, RNA-seq, scRNA-seq, and Drug Database Resources Used for Hypothyroidism and Cirrhosis Research.**
(XLSX)

**S2 Table. LDSC Results with Hypothyroidism and Cirrhosis.**
(XLSX)

**S3 Table. Screening of Genetic Instruments for Potential Horizontal Pleiotropy via GWAS Catalog.**
(XLSX)

**S4 Table. SNPs Used for MR Analysis.**
(XLSX)

**S5 Table. MR and Meta Analysis Results with Hypothyroidism and Cirrhosis.**
(XLSX)

**S6 Table. TWAS Results within the OTTERS Framework for Identifying Genes Associated with Hypothyroidism.**
(XLSX)

**S7 Table. TWAS Results within the OTTERS Framework for Identifying Genes Associated with Cirrhosis.**
(XLSX)

**S8 Table. Results of the UpSet plot for the different Gene Sets.**
(XLSX)

**S9 Table. Intersection of GO and KEGG Enrichment Results.**
(XLSX)

**S10 Table. Genes Significantly Affected After Virtual Knockout.**
(XLSX)

**S11 Table. Significantly Affected Pathways After Virtual Knockout of Key Genes.**
(XLSX)

**S12 Table. Molecular Docking Identifies Thyroid and Hepatobiliary Compounds Targeting with Hub Genes.**
(XLSX)

**S13 Table. 675 Diseases from UK Biobank.**
(XLSX)

**S14 Table. Results of Hub Genes Intervention on-disease Identified through PheW-MR Analysis.**
(XLSX)

**S15 Table. Synthesis of Trait-Specific Genetic Effects into an Overall Effect per Gene.**
(XLSX)

**S1 Checklist. STROBE-MR checklist.**
(DOCX)

**S1 Fig. Study design for MR (Created in BioRender. L, E. (2025)** https://BioRender.com/484biwe).
(TIFF)

**S1 Text. Supplementary methods for WGCNA and PheW-MR.**
(DOCX)

## Acknowledgments

The authors would like to express their sincere gratitude to Rui Wang, MS, from the Department of Biostatistics at the University of Washington, for his valuable guidance on statistical methodology, which greatly contributed to the rigor and robustness of the analytical framework. We also gratefully acknowledge the Asian Pacific Association for the Study of the Liver (APASL) for the recognition of our work at the APASL 2025 Congress (Submission ID: SA-APASL2025-15672).

## Author contributions

**Conceptualization:** Ziyang Yang, Weixuan Liang, Qi Zhang.

**Data curation:** Ziyang Yang, Weixuan Liang, Qi Zhang, Can Weng, Zhuofeng Wen.

**Formal analysis:** Hao Deng.

**Funding acquisition:** Jiyuan Zhou.

**Investigation:** Ziyang Yang, Weixuan Liang, Can Weng, Hao Deng, Jingwen Deng.

**Methodology:** Ziyang Yang, Weixuan Liang, Qi Zhang, Can Weng.

**Project administration:** Ziyang Yang, Weixuan Liang, Qi Zhang.

**Resources:** Ziyang Yang, Hui Yang, Jiyuan Zhou.

**Supervision:** Hui Yang, Jiyuan Zhou.

**Validation:** Ziyang Yang, Weixuan Liang, Qi Zhang, Zhixin Xie, Yiwei Lin.

**Visualization:** Zhixin Xie, Yiwei Lin.

**Writing – original draft:** Ziyang Yang, Weixuan Liang, Qi Zhang, Jingyi Wu.

**Writing – review & editing:** Xiuling Fu, Chengxin Gu, Tao Yang, Hui Yang, Jiyuan Zhou.

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
