## [Decision Letter · Decision Letter 0]

11 Aug 2025

PGENETICS-D-25-00597

The role of hypothyroidism in cirrhosis pathogenesis: a retrospective cohort study and multi-omics integration analysis

PLOS Genetics

Dear Dr. Zhou,

Thank you for submitting your manuscript to PLOS Genetics. After careful consideration, we feel that it has merit but does not fully meet PLOS Genetics's publication criteria as it currently stands. Therefore, we invite you to submit a revised version of the manuscript that addresses the points raised during the review process.

Please submit your revised manuscript within 60 days Oct 10 2025 11:59PM. If you will need more time than this to complete your revisions, please reply to this message or contact the journal office at plosgenetics@plos.org. Please include the following items when submitting your revised manuscript:

We look forward to receiving your revised manuscript.

Kind regards,

Renato Polimanti, Ph.D.

Academic Editor

PLOS Genetics

Zoltán Kutalik

Section Editor

PLOS Genetics

Aimée Dudley

Editor-in-Chief

PLOS Genetics

Anne Goriely

Editor-in-Chief

PLOS Genetics

**Journal Requirements:**

**Reviewers' comments:**

Reviewer's Responses to Questions

Reviewer #1: Dr Zhou and colleagues investigated the genetic overlap between hypothyroidism and liver cirrhosis using publicly available genome-wide association study (GWAS) summary statistics from large biobanks such as UK Biobank and FinnGen. The authors report a significant observational association between the two diseases based on MIMIC-IV data, a significant genetic correlation between published GWAS of the two diseases using LD score regression, and a significant causal effect of hypothyroidism on cirrhosis risk (but no evidence of a causal effect of cirrhosis on hypothyroidism risk) which was replicated in an independent cohort. The authors then incorporated genomic data from multiple databases which highlighted HLA-DQA1 and CD27 as key molecules linking hypothyroidism and liver cirrhosis.

Thank you to the authors for sharing this interesting study. Overall, I think the manuscript is clear and well written. I have no major concerns at this time. However, I invite the authors to address the following minor comments/suggestions to improve clarity:

Line 151-153: A lot of individuals were excluded from the retrospective analysis due to missing data (~36% of cirrhosis cases and ~66% of cirrhosis controls excluded). What are the reasons for this level of missing data? Please discuss whether differences in the missingness between cases and controls could have given rise to the observational associations reported.

Line 159-165: Are there any other important covariates that may act as confounders for these associations – smoking, BMI, medication?

Line 185-187: The genetic correlation is technically statistically significant, but the effect estimate is fairly weak (rg = 0.1566). I would suggest adding some text to the discussion to highlight the weak magnitude of effect.

Line 187-190: Can you comment on how these SNP heritabilities compare to estimates from twin studies for hypothyroidism and liver cirrhosis (where possible)? Is it accurate to describe a SNP heritability of 4.56% as indicating genetic factors play a “prominent role” in hypothyroidism? Are you able to make comparisons to SNP heritabilities based on common variant GWAS for other relevant disease traits?

Line 200: Is it correct to refer to this as “progression of cirrhosis”? Line 208 instead refers to the outcome as cirrhosis risk. If I understood correctly, it’s later in the paper you include hepatitis as an intermediate progression trait.

Line 240-243: It’s not clear to me what is meant by a “gene module”. I think this requires further explanation before discussing the results from WGCNA. In addition, I suggest adding some context for whether a network of 4393 genes for the hypothyroidism analysis is meaningful given that this would represent a significant proportion of all genes in the human genome. Similarly for cirrhosis. In particular, is it surprising to find significant phenotypic correlation given the number of genes reported?

General: How sensitive are the multi-omics approaches to the GWAS summary statistics used? Given that this was based on common variant GWAS, how much would be missing by not including rare variants and more complex genetic variation such as structural variants in the input to these methods?

Figure 3B+D: I suggest having both “normal” columns on the same side of each plot to help clarity.

Figure 4A: I think justification could be given for not having looked at genes that were supported by fewer than 5 criteria. For example, there are 22 genes that were supported by DEG_HT, DEG_LC, WGCNA_HT and WGCNA_LC (not TWAS). Since the TWAS did not, individually, provide evidence for many genes relative to the other methods, there could be some important genes that were missed by the approach taken.

PheW-MR analysis: Certain disease traits may be better studied using quantitative measures for a more powerful analysis (e.g. lung function measures such as FEV1 and FVC instead of dichotomising into cases/controls for chronic obstructive pulmonary disease). Were you able to consider quantitative measures in this analysis? If so, why did you exclude them?

Supplementary Table 11: I think it would be beneficial to the include associations between HLA-DQ1 and all 675 disease traits, not just those that were statistically significant. Maybe the significant results can be highlighted in some way to make it obvious that they were, indeed, statistically significant. I would also advocate for including the CD27 results for completeness.

Reviewer #2: This manuscript presents a comprehensive assessment of the relationship between hypothyroidism and liver cirrhosis. The study integrates both traditional observational (retrospective cohort) and multi-omics approaches. The use of diverse analytic methods is ambitious and methodologically rich. The identification of HLA-DQA1 and CD27 as immune-related mediators is biologically plausible and translationally relevant. However, several areas require clarification to improve reproducibility and interpretability. Here are some comments to help improve the manuscript:

1. The Introduction provides a strong rationale but lacks explicitly stated research questions or objectives. Authors need to add concise sentences at the end of the Introduction section outlining the study-specific objectives.

2. The Methods section describes cohort selection, inclusion/exclusion criteria, covariates, and logistic regression modelling. In the Results, authors need to briefly mention (in-text, as Table 1 already notes the variables) the covariates adjusted for in the multivariate analysis to aid reader clarity.

3. Figure 1 looks good; however, it will be helpful to provide a figure that captures all the study approaches and shows how the analysis progresses from one level to another (i.e., analytic workflow).

4. Line 183: authors mention “To further explore potential genetic underpinnings of…” This is the first method applied for genetic assessment in the study, so it is unclear what “further” refers to. Also, please clarify the scale used for the SNP-based heritability estimates; was it on the observed or liability scale?

5. For the MR analysis, a few issues remain to be clarified:

- The STROBE-MR checklist information and page numbers provided do not correspond to the content in the manuscript or are missing.

- In line with reporting guidelines, it is important to include a schematic diagram summarising the MR design and clearly showing the core assumptions of the method.

- The authors mention using the GWAS Catalog and PhenoScanner V2 to assess exclusion restriction. Please describe how this was done. Importantly, note that PhenoScanner has been offline for an extended period (and the command line version does not appear to work), so clarification is needed here.

6. Authors performed TWAS using OTTERS. It would be helpful to briefly justify the use of OTTERS over other TWAS tools and to discuss the rationale for tissue choice. In particular, please reflect on the implications of using blood-based eQTLs for diseases such as cirrhosis and hypothyroidism.

7. The manuscript acknowledges the use of Hashimoto’s thyroiditis data as a proxy and discusses its limitations. Authors may wish to briefly expand on how this may bias immune-related findings in the differential expression analysis.

8. Replication appears to be limited to the cirrhosis component of the analysis. Could there also have been replication for hypothyroidism, or is there a rationale for the current approach? A brief discussion would be helpful.

9. Authors need to correct typographic errors (e.g., ‘gens’ should be ‘genes’). Please check for similar issues throughout the manuscript and correct them accordingly.

10. Authors need to read through the manuscript line by line to ensure there are no grammatical errors or incomplete statements. For example, in line 112: ‘Building on this…’. it is not clear what ‘this’ refers to. Please avoid such unclear referencing throughout the manuscript.

11. The Discussion section could be streamlined to reduce redundancy. There are several places where results are restated unnecessarily. Consider focusing more on interpretation and implications.

12. It appears that the manuscript may have originally been structured with Methods before Results/Discussion. Please go through the current version and ensure that the narrative progresses coherently from Introduction to Results, then Discussion, and that abbreviations used in the Results are defined at first use. For example, some abbreviations appear in the Results but are only defined later in the Methods section.

**Have all data underlying the figures and results presented in the manuscript been provided?**

Reviewer #1: None

Reviewer #2: **No: ** Publicly available data were analysed. Authors provided references and links to the data in the Supplementary Table.

PLOS authors have the option to publish the peer review history of their article (what does this mean? ). If published, this will include your full peer review and any attached files.

**Do you want your identity to be public for this peer review?** For information about this choice, including consent withdrawal, please see our Privacy Policy .

Reviewer #1: No

Reviewer #2: **Yes: ** Emmanuel Adewuyi

**Figure resubmission:**
---

## [Decision Letter · Decision Letter 1]

21 Oct 2025

PGENETICS-D-25-00597R1

The role of hypothyroidism in cirrhosis pathogenesis: a retrospective cohort study and multi-omics integration analysis

PLOS Genetics

Dear Dr. Zhou,

Thank you for submitting your manuscript to PLOS Genetics. After careful consideration, we feel that it has merit but does not fully meet PLOS Genetics's publication criteria as it currently stands. Therefore, we invite you to submit a revised version of the manuscript that addresses the points raised during the review process.

Please submit your revised manuscript within 30 days Nov 20 2025 11:59PM. If you will need more time than this to complete your revisions, please reply to this message or contact the journal office at plosgenetics@plos.org. Please include the following items when submitting your revised manuscript:

We look forward to receiving your revised manuscript.

Kind regards,

Renato Polimanti, Ph.D.

Academic Editor

PLOS Genetics

Zoltán Kutalik

Section Editor

PLOS Genetics

Aimée Dudley

Editor-in-Chief

PLOS Genetics

Anne Goriely

Editor-in-Chief

PLOS Genetics

**Reviewers' comments:**

Reviewer's Responses to Questions

Reviewer #1: Thank you to Dr Zhou and colleagues for addressing my prior concerns. I have no further suggestions to make.

Reviewer #2: Thank you for responding to my comments. I have two minor comments.

1. In response to one of the reviewer 1's comments, authors stated that their revised analysis is “more robust” because the SNP-based heritability was converted to the liability scale, which they suggested influences the genetic correlation (rg). However, the liability-scale adjustment applies to heritability estimates (h²) rather than to rg. Thus, converting to the liability scale does not improve or correct rg estimation. If the updated rg (0.2135) differs from the preliminary value (0.1566), it likely reflects differences in input data, QC parameters, or model specification rather than the liability transformation. Additionally, if gencov intercept is constrained, it could also affect rg estimates, especially when sample overlap is not fully accounted for. The authors should clarify what they did here.

2. There was heterogeneity in the MR analysis, which would suggest horizontal pleiotropy, even with pleiotropy test suggesting otherwise (a common scenario in MR). To address this situation, authors need to check through their instruments and remove any of them associated with the outcome variable at p < 0.05. Lets see what outcome you have thereafter, but I would expect at least the heterogeneity would be gone. Please, also report the p values, both for outcome and exposure, for all the instruments used (probably as a supplementary table) for transparency.

**Have all data underlying the figures and results presented in the manuscript been provided?**

Reviewer #1: Yes

Reviewer #2: Yes

PLOS authors have the option to publish the peer review history of their article (what does this mean? ). If published, this will include your full peer review and any attached files.

**Do you want your identity to be public for this peer review?** For information about this choice, including consent withdrawal, please see our Privacy Policy .

Reviewer #1: No

Reviewer #2: **Yes: ** Emmanuel Adewuyi

**Figure resubmission:**
---

## [Editor Report · Decision Letter 2]

31 Oct 2025

Dear Dr Zhou,

We are pleased to inform you that your manuscript entitled "The role of hypothyroidism in cirrhosis pathogenesis: a retrospective cohort study and multi-omics integration analysis" has been editorially accepted for publication in PLOS Genetics. Congratulations!

Yours sincerely,

Renato Polimanti, Ph.D.

Academic Editor

PLOS Genetics

Zoltán Kutalik

Section Editor

PLOS Genetics

Aimée Dudley

Editor-in-Chief

PLOS Genetics

Anne Goriely

Editor-in-Chief

PLOS Genetics

BlueSky: @plos.bsky.social

Comments from the reviewers (if applicable):

**Data Deposition**

http://datadryad.org/submit?journalID=pgenetics&manu=PGENETICS-D-25-00597R2

**Press Queries**

---

## [Editor Report · Acceptance letter]

PGENETICS-D-25-00597R2

The role of hypothyroidism in cirrhosis pathogenesis: a retrospective cohort study and multi-omics integration analysis

Dear Dr Zhou,

We are pleased to inform you that your manuscript entitled "The role of hypothyroidism in cirrhosis pathogenesis: a retrospective cohort study and multi-omics integration analysis" has been formally accepted for publication in PLOS Genetics! Your manuscript is now with our production department and you will be notified of the publication date in due course.

With kind regards,

Zsofia Freund

PLOS Genetics

On behalf of:
